# Distribution, epidemiology, and antimicrobial resistance pattern of gram-negative bacteria isolated from blood: A retrospective study in a tertiary care hospital, Dhaka, Bangladesh

Nusrat Noor Tanni*, Maherun Nesa‡, Rubaiya Binte Kabir‡, Farjana Binte Habib‡, Md. Asaduzzaman‡, Avizit Sarker‡, Md. Faizur Rahman‡, Noor E. Jannat Tania‡, Azmeri Haque‡, Umme Saoda‡, Kakali Halder‡, Nadira Akter‡, Rozina Aktar Zahan‡, Mahbuba Chowdhury‡, Sazzad Bin Shahid‡

Department of Microbiology, Dhaka Medical College, Shahbag, Dhaka, Bangladesh

‡ MN, RBK, FBH, MC and SBS are contributed equally in conceptualization, data collection, statistical analysis and manuscript writing. MA, AS, MFR, NEJT, AH, US, KH, NA and RAZ are contributed equally in data collection and manuscript editing.
* nusratnoortanni@gmail.com

## Abstract

The effective treatment of bloodstream infections caused by gram-negative bacteria (GNB) poses significant challenges as their distribution and resistance patterns vary across geographic locations and healthcare settings. Data on bloodstream infections (BSIs) and antimicrobial resistance patterns (AMR) in Bangladesh are limited. The objective of this study was to address this challenge by analyzing the prevalence, distribution, and resistance patterns—including multidrug-resistant (MDR) and extensively drug-resistant (XDR) status—of Gram-negative bacteria isolated from blood cultures at Dhaka Medical College Hospital, stratified by age, sex, and hospital unit. This retrospective study was conducted between November 2023 to October 2024 in the Department of Microbiology, Dhaka Medical College, Bangladesh. Bacterial blood culture and susceptibility testing records of GNB from both the inpatient department (IPD), intensive care units (ICU), and outpatient department (OPD) samples, irrespective of age and sex, were included and analyzed in this study. Total 3753 blood samples were analyzed in this study period, among them 427 blood samples yield bacterial growth. Out of 427 isolates, gram-negative bacteria were 87.6%, with a slightly higher prevalence in male patients (57.2%). *Salmonella* spp was the most prevalent isolate from the OPD, while *Acinetobacter* spp was predominant in IPD and ICU. The highest antimicrobial resistance was observed to ceftazidime in all isolated GNB, except *Salmonella* spp. *Acinetobacter* spp was predominantly multidrug-resistant (MDR) (75.4%), and the lowest was *Salmonella* spp (40.7%). Among 15% extensively drug-resistant (XDR) isolates, the majority were *Acinetobacter* spp, followed by *Pseudomonas* spp and *Klebsiella* spp. The highest prevalence of both

**Data availability statement:** All the data related to this manuscript have been uploaded as Supporting Information files.

**Funding:** The author(s) received no specific funding for this work.

**Competing interests:** The authors have declared that no competing interests exist.

MDR and XDR organisms were observed in the ICU. The antibiotic resistance trends display restricted effectiveness of commonly used antibiotics, such as cephalosporins and fluoroquinolones, compelling dependence on last-resort antibiotics- colistin. Systematic local surveillance and epidemiological studies of antimicrobial resistance would assist in taking measures to slow down the spread of resistance.

## Introduction

Bloodstream infection (BSI) is one of the most important causes of hospitalization and mortality globally, and its clinical presentation ranges from transient asymptomatic bacteremia to life-threatening septicemia and septic shock [1,2]. Although any age group can be affected, infants and children are at the highest risk of contracting such infections [2]. Rapid and reliable bacteremia diagnosis entails aseptic blood collection for culture and sensitivity testing before the administration of antimicrobial agents [3]. Gram-negative bacteria (GNB) were considered the cause of around 25% of nosocomial bacteremia and 45% of community-acquired bacteremia [4]. Based on the current population, BSI caused by GNB accounts for 279,000 cases and 33,500–41,900 deaths annually in the USA [5]. Gram-negative pathogens exhibit a higher case fatality rate in neonates with sepsis than Gram-positive pathogens (59% vs. 33%) [6]. Most notably isolated GNBs are *Escherichia coli, Klebsiella pneumoniae, Enterobacter cloaca, Salmonella typhi, Pseudomonas aeruginosa,* and *Acinetobacter baumannii* [7]. Bacterial distribution and sensitivity trends vary across geographic locations and healthcare facilities [8]. Antibiotic-resistant strains, particularly those of Gram-negative bacteria, are emerging at an alarming rate, posing significant challenges to the effective treatment of BSI [9].

Antimicrobial resistance (AMR) has emerged as a critical global health crisis. Infections caused by multidrug-resistant (MDR) bacteria significantly increase morbidity and mortality, leading to extended hospital stays, consumption of more expensive antibiotics, and the risk of developing antimicrobial resistance. Consequently, MDR infections not only threaten patient outcomes but also result in substantial financial losses for healthcare systems [10].

AMR surveillance plays a pivotal role in combating and managing this crisis, as emphasized in the World Health Organization's (WHO) *Global Strategy for Containment of Antimicrobial Resistance* (2001). Furthermore, adopting the *Global Action Plan on Antimicrobial Resistance* (GAP) by the 68th World Health Assembly in May 2015 underscores the importance of ensuring the sustained efficacy of antimicrobials for treating and preventing infectious diseases [11]. Understanding the epidemiology of Gram-negative BSIs and AMR trends is vital for selecting empirical antibiotics and optimizing antibiotic therapy regimens [12].

Since every geographical area has a unique pattern of resistant organisms, monitoring local resistant patterns consistently can guide the appropriate use of antimicrobial agents and contribute to preventing antimicrobial resistance. Although the mainstream data on bacterial pathogen surveillance and AMR profile comes

from high-income nations, it is well acknowledged that AMR is a worldwide concern that mostly affects low-income countries. Scrutiny of microorganism distribution, BSI epidemiology, changing antibiotic resistance rates, and demographics are required to support appropriate therapy. This study aimed to ascertain the prevalence, distribution, and antimicrobial resistance patterns with MDR, and XDR status, of commonly isolated GNB by age, sex, and hospital units through the analysis of the blood cultures of patients at Dhaka Medical College Hospital (DMCH), a tertiary care hospital in Dhaka, Bangladesh.

## Materials and methods

This was a retrospective study of records for bacterial culture and susceptibility testing results between November 2023 to October 2024 at the Department of Microbiology, Dhaka Medical College, Dhaka, Bangladesh (S1 Data). Microbiology laboratory data regarding gram-negative bacterial growth and antimicrobial susceptibility testing results were accessed on March 05, 2025.

Among culture-positive blood samples, only gram-negative bacteria culture records were included and analyzed in this study. Both inpatient and outpatient samples, regardless of age or hospital setting, were included. All laboratory data of blood culture samples with contamination and repetitive isolated species of the same patient in the same specimen type were excluded from this present study; only the first isolate was considered. The following variables were included: age, gender, hospital units, name of the organisms, antimicrobial disk used for susceptibility testing, and susceptibility results of each antibiotic tested.

### Ethics statement

Ethical approval for the study was obtained from the Institutional Review Board (IRB) of Dhaka Medical College (memo no: IRB-DMC/2025/25). IRB have waived the requirement of informed consent for this retrospective study of laboratory data records. All the data were anonymized by assigning an identification number to each participant. However, data confidentiality and the right to privacy were preserved per the Declaration of Helsinki.

### Blood collection

With strict aseptic technique, 10 ml of venous blood was collected from adult patients, and 3–4 ml from pediatric patients. The blood samples were inoculated immediately into an automated blood culture bottle and analyzed by an automated monitoring system for bacterial detection (BacT/Alert 3D 60, BioMerieux, France), incubated at 37°C aerobically until a positive culture was observed or up to a maximum of 7 days [13]. In some instances, when automated blood culture bottles were not available, a few blood samples were inoculated in conventional blood culture bottles containing Trypticase soy Broth (TSB), incubated at 37°C aerobically overnight, and examined daily for turbidity up to 7 days.

### Microbial identification

Subculture was carried out on the MacConkey agar and Blood agar plates, incubated at 35–37 °C overnight, and then examined for visible growth. From the colony, gram-negative bacteria were identified by Gram staining, biochemical tests including the oxidase test, Triple Sugar Iron (TSI), Motility Indole Urea (MIU), and citrate tests. In addition, Gram-positive organisms were identified using the catalase test and coagulase test [14]. The Antibiotic susceptibility testing was performed by the Kirby-Bauer disk diffusion method, following laboratory protocol, and the interpretation of the zone of inhibition was done according to Clinical and Laboratory Standards Institute (CLSI), 2022 guidelines [15]. Antibiotic susceptibility was done using following discs- Amoxicillin-clavulanic acid (20/10µg), Piperacillin-Tazobactum (100/10µg), Ceftriaxone (30µg), Ceftazidime (30µg), Cefixime (5µg), Cefepime (30µg), Meropenem (10µg), Ciprofloxacin (5µg), Gentamicin (10µg), Amikacin (30µg), Netilmicin (30µg), Tigecycline (15µg), Doxycycline (30µg),

Trimethoprim-Sulphamethoxazole (1.25/23.75µg), and Azithromycin (15µg) disk. Susceptibility to tigecycline was defined according to the Food and Drug Administration (FDA), Identified Interpretive Criteria 2023; isolates were categorized as resistant if the zone diameter was < 14 mm and susceptible if the zone diameter was ≥ 19 mm [16].

### Statistical analysis

Data for categorical variables were presented as numbers and percentages, and mean ± standard deviation (SD) or as the median for continuous data. The test used for a normal distribution was the Shapiro–Wilk test. The Pearson Chi-square test and Fisher's exact test were used to define statistical significance. All tests with *p-value* < 0.05 were considered significant with a 95% confidence interval. The data were statistically evaluated with Statistical Package for the Social Sciences (SPSS) version 23.0 (IBM Corp., Armonk, NY).

Operational definition of multidrug-resistant (MDR) and extensively drug-resistant (XDR) Bacteria: [17]

MDR is defined as non-susceptibility to at least one agent in three or more antimicrobial categories. XDR is defined as non-susceptibility to at least one agent in all but two or fewer antimicrobial categories (i.e., bacterial isolates remain susceptible to only one or two categories).

### Results

A total of 3753 blood samples were analyzed in this study period amid 427 blood samples yield bacterial growth and 3326 didn't yield growth of bacteria. Among these 427 isolates, gram-positive organisms were 53 (12.4%), among them *S. aureus* was 90.6%, *Coagulase Negative Staphylococcus* 7.5%, and *Enterococcus* spp. was 1.9%. The total number of isolated gram-negative bacteria was 374 (87.6%). Isolation of GNB was slightly higher in male patients 214 (57.2%) compared to female 160 (42.8%), the male-female ratio was 1.33:1.

The most common isolated GNB was *Salmonella spp.* 189 (50.6%), followed by *Acinetobacter* spp 69 (18.5%), *Pseudomonas* spp 46 (12.3%), *Klebsiella* spp 33 (8.8%), *Escherichia coli* (*E. coli*) 21 (5.6%), *Enterobacter* spp 12 (3.2%), *Citrobacter* spp 2 (0.5%), and *Proteus* spp 2 (0.5%).

The relationship between organisms isolated in the blood sample and the gender of patients is presented in Table 1. Notably, no significant association between males and females in terms of isolates was observed (p = 0.177). Among the isolates, *Salmonella* spp was the most prevalent bacteria, followed by *Acinetobacter* spp in both female and male groups.

The age of patients ranged from <1 year to 82 years with a mean age of 27.10 years and a standard deviation of 19.003 years. Between the females and males, the mean age ± SD was 27.53 ± 20.35, and 26.78 ± 17.98 years, and the median was 22.00 and 23.00 years respectively. Shapiro–Wilk test for normal distribution was done, which rejects

**Table 1. Distribution of GNB according to gender (N = 374).**

| Organisms | Total N (%) 374 (100) | Female N (%) 160 (42.8) | Male N (%) 214 (57.2) | p-value (test type) |
|---|---|---|---|---|
| *Salmonella* spp | 189 (50.6) | 70 (37.0) | 119 (63.0) | 0.177 (C) |
| *Acinetobacter* spp | 69 (18.5) | 33 (47.8) | 36 (52.2) | |
| *Pseudomonas* spp | 46 (12.3) | 21 (45.7) | 25 (54.3) | |
| *Klebsiella* spp | 33 (8.8) | 15 (45.5) | 18 (54.5) | |
| *E. coli* | 21 (5.6) | 14 (66.7) | 7 (33.3) | |
| *Enterobacter* spp | 12 (3.2) | 6 (50.0) | 6 (50.0) | |
| *Citrobacter* spp | 2 (0.5) | 0 (0) | 2 (100.0) | |
| *Proteus* spp | 2 (0.5) | 1 (50.0) | 1 (50.0) | |

C- Chi-Square test.

normality (*p*-value < 0.05). The age range was distributed into six intervals to individualize the possible relationship between isolates and age (Table 2). In age interval analysis, *Acinetobacter* spp, *Klebsiella* spp, and *Pseudomonas* spp were more prevalent in the ≤ 18 years. Among 19–30 years age group more *Salmonella* spp were isolated.

The association between the isolates and hospital settings were analyzed (Table 3); Hospital units were assigned into Outpatient departments (OPD), Inpatient Departments (IPD), and Intensive Care Units (ICU).

Antimicrobial resistance patterns for various gram-negative bacterial species against different categories of antimicrobials were presented in Table 4. *E. coli* showed the highest resistance to ciprofloxacin (95.2%), followed by amoxicillin-clavulanic acid, piperacillin-tazobactam, ceftriaxone, and ceftazidime (90.5% each). *Klebsiella* spp exhibited 100% resistance to ceftazidime, followed by ceftriaxone and piperacillin-tazobactam (97%), with low resistance to netilmicin (54.5%), and tigecycline (60.6%). *Acinetobacter* spp and *Pseudomonas* spp demonstrated frequent resistance to third-generation cephalosporins. For *Salmonella* spp, most isolates were resistant to ciprofloxacin (95.2%).

Distribution of isolated organisms and hospital units according to drug resistance status were evaluated in Table 5, which lists the number of resistant isolates per total tested. Total MDR was 53.7%, the highest MDR positive rate

**Table 2. Isolated Gram-negative bacteria stratified by age intervals (N = 374).**

| Isolated Organisms | Age Intervals N (%) | | | | | |
|---|---|---|---|---|---|---|
| | ≤ 18 years | 19-30 years | 31- 40 years | 41-50 years | 51- 60 years | ≥ 61 years |
| *Salmonella* spp. (189) | 77 (40.7) | 93 (49.2) | 12 (6.3) | 1 (0.5) | 2 (1.1) | 4 (2.2) |
| *Acinetobacter* spp (69) | 18 (26.1) | 18 (26.1) | 10 (14.5) | 9 (13.0) | 8 (11.6) | 6 (8.7) |
| *Pseudomonas* spp (46) | 10 (21.7) | 9 (19.6) | 8 (17.4) | 5 (10.8) | 9 (19.6) | 5 (10.9) |
| *Klebsiella* spp (33) | 9 (27.2) | 4 (12.1) | 6 (18.2) | 5 (15.2) | 5 (15.2) | 4 (12.1) |
| *E. coli* (21) | 6 (28.6) | 4 (19.0) | 1 (4.8) | 5 (23.8) | 0 (0.0) | 5 (23.8) |
| *Enterobacter* spp (12) | 4 (33.3) | 2 (16.7) | 1 (8.3) | 3 (25.0) | 2 (16.7) | 0 (0.0) |
| *Citrobacter* spp (2) | 2 (100.0) | 0 (0.0) | 0 (0.0) | 0 (0.0) | 0 (0.0) | 0 (0.0) |
| *Proteus* spp (2) | 0 (0.0) | 0 (0.0) | 0 (0.0) | 0 (0.0) | 1 (50.0) | 1 (50.0) |
| Total | 126 (33.7) | 130 (34.7) | 38 (10.2) | 28 (7.5) | 27 (7.2) | 25 (6.7) |

**Table 3. Distribution of Gram-negative bacteria in different hospital settings.**

| Organisms (374) | Hospital Settings N (%) | | | *p*-value (test type) |
|---|---|---|---|---|
| | OPD 212 (56.7) | IPD 142 (38.0) | ICU 20 (5.3) | |
| *Salmonella* spp (189) | 163 (76.8) | 24 (16.9) | 2 (10.0) | p < 0.001 (C) |
| *Acinetobacter* spp (69) | 13 (6.1) | 47 (33.1) | 9 (45.0) | |
| *Pseudomonas* spp (46) | 12 (5.7) | 31 (21.8) | 3 (15.0) | |
| *Klebsiella* spp (33) | 12 (5.7) | 20 (14.1) | 1(5.0) | |
| *E. coli* (21) | 7 (3.3) | 13 (9.2) | 1 (5.0) | |
| *Enterobacter* spp (12) | 5 (2.4) | 5 (3.5) | 2 (10.0) | |
| *Citrobacter* spp (2) | 0 (0.0) | 2 (1.4) | 0 (0.0) | |
| *Proteus* spp (2) | 0 (0.0) | 0 (0.0) | 2 (10.0) | |

C- Chi-Square test.

There was a significant association between hospital units and isolated organisms (*p* < 0.001).

*Salmonella* spp (76.8%) was the most frequent isolate in the OPD, while *Acinetobacter* spp (45.0%, 33.1%) followed by *Pseudomonas* spp (15.0%, 21.8%), were most frequent in ICU and IPD accordingly.

**Table 4. Antimicrobial resistance pattern of isolated Gram-Negative Bacteria.**

| Organisms (N) | Antibiotic Categories | | | | | | | | | | | | | | |
| | Penicillin | | Cephalosporin third gen | | | Cephalosporin fourth gen | Carbapenems | Fluoroquinolones | Aminoglycoside | | | Glycylcyclines | Tetracycline | Folate pathway inhibitors | Macrolides |
| | AMC N (%) | TZP N (%) | CRO N (%) | CAZ N (%) | CFM N (%) | FEP N (%) | MEM N (%) | CIP N (%) | GEN N (%) | AMK N (%) | NET N (%) | TGC N (%) | DOX N (%) | SXT N (%) | AZM N (%) |
|---|---|---|---|---|---|---|---|---|---|---|---|---|---|---|---|
| *Acinetobacter* spp (69) | – | 65 (94.2) | 67 (97.1) | 68 (98.5) | 67 (97.1) | – | 44 (63.8) | 52 (75.4) | 55 (79.7) | 47 (68.1) | 53 (76.8) | 42 (60.8) | 44 (63.8) | 56 (81.1) | |
| *Pseudomonas* spp (46) | – | 35 (76.1) | – | 38 (82.6) | – | 37 (80.4) | 28 (60.9) | 31 (67.4) | 32 (69.6) | 30 (65.2) | 23 (50) | – | – | – | |
| *Klebsiella* spp (33) | 27 (81.8) | 32 (97) | 32 (97) | 33 (100) | 31 (94.6) | – | 24 (72.7) | 32 (97) | 24 (72.7) | 26 (78.8) | 18 (54.5) | 20 (60.6) | 20 (60.6) | 28 (84.8) | |
| *E. coli* (21) | 19 (90.4) | 19 (90.5) | 19 (90.5) | 19 (90.5) | 18 (85.7) | – | 11 (52.4) | 20 (95.2) | 15 (71.4) | 17 (80.9) | 13 (61.9) | 9 (42.8) | 16 (76.2) | 16 (76.2) | |
| *Enterobacter* spp (12) | 9 (75.0) | 11 (91.7) | 12 (100) | 12 (100) | 12 (100) | – | 7 (58.3) | 10 (83.3) | 8 (66.7) | 8 (66.7) | 4 (33.3) | 8 (66.7) | 7 (58.3) | 9 (83.3) | |
| *Citrobacter* spp (2) | 2 (100) | 2 (100) | 1 (50) | 2 (100) | 2 (100) | – | 2 (100) | 1 (50) | 2 (100) | 2 (100) | 2 (100) | 0 (0) | 1 (50) | 1 (50) | |
| *Proteus* spp (2) | 2 (100) | 1 (50) | 2 (100) | 2 (100) | 2 (100) | – | 1 (50) | 2 (100) | 2 (100) | 2 (100) | 0 (100) | 2 (100) | 1 (50) | 2 (100) | |
| *Salmonella* spp (189) | 63 (33.3) | – | 23 (12.1) | – | 79 (41.8) | 57 (30.1) | – | 180 (95.2) | – | – | – | – | – | 59 (31.2) | 68 (36) |
| Total (374) | 122/259 (47.1) | 165/185 (89.2) | 157/328 (47.8) | 174/185 (94.1) | 211/328 (64.3) | 94/235 (40.0) | 117/185 (63.2) | 328/374 (87.7) | 138/185 (74.6) | 132/185 (71.3) | 111/185 (60.0) | 81/139 (58.2) | 89/139 (64.0) | 171/328 (52.1) | 68/189 (36) |

Abbreviations: AMC- Amoxicillin/ clavulanic acid, TZP- Piperacillin/Tazobactum, CRO- Ceftriaxone, CAZ- Ceftazidime, CFM- Cefixime, FEP- Cefepime, MEM- Meropenem, CIP- Ciprofloxacin, GEN- Gentamicin, AMK- Amikacin, NET- Netilmicin, TGC- Tigecycline, DOX- Doxycycline, SXT- Trimethoprim/Sulphamethoxazole, AZM- Azithromycin.

**Table 5. Distribution of isolated organisms and hospital settings according to their drug resistance status (N = 374).**

| | Drug resistance status N (%) | |
| --- | --- | --- |
| | MDR 201 (53.7) | XDR 56 (15) |
| **Organisms** | | |
| *Salmonella* spp (189) | 77/189 (40.7) | 0/189 (0) |
| *Acinetobacter spp* (69) | 52/69 (75.4) | 29/69 (42.0) |
| *Pseudomonas* spp (46) | 23/46 (50.0) | 12/46 (26.1) |
| *Klebsiella* spp (33) | 23/33 (69.7) | 7/33(21.2) |
| *E. coli* (21) | 15/21 (71.4) | 4/21 (19.0) |
| *Enterobacter* spp (12) | 8/12 (66.7) | 4/12 (33.3) |
| *Citrobacter* spp (2) | 2/2 (100.0) | 0/2 (0.0) |
| *Proteus* spp (2) | 1/2 (50.0) | 0/2 (0.0) |
| *p-value* (test type) | <0.000 (F) | <0.000 (F) |
| **Hospital Settings** | | |
| ICU (20) | 16/20 (80.0) | 10/20 (50.0) |
| IPD (142) | 94/142 (66.1) | 35/142 (24.6) |
| OPD (212) | 91/212 (42.9) | 11/212 (5.2) |
| p-value (test type) | <0.000 (C) | <0.000 (C) |

F- Fisher exact test, C- Chi-Square test.

observed for *Citrobacter* spp (100.0%), though the sample size was very small (only two isolates). The lowest MDR-positive organisms were *Salmonella* spp (40.7%). There is a statistically significant association (p < 0.000) between the type of organisms and MDR status. About 15% of all isolated GNBs were identified as XDR as per the definition. Among the *Acinetobacter* spp, 42.5% were XDR, followed by *Pseudomonas* spp (26.1%) and *Klebsiella* spp (21.1%). Notably, no XDR isolates were detected among *Salmonella* spp in this current study. When associating resistance across hospital settings, the highest prevalence of both MDR and XDR was observed in the intensive care unit (ICU), with rates of 80.0% and 50.0%, respectively. In contrast, the OPD recorded the lowest prevalence, with 42.9% MDR and 5.2% XDR isolates.

## Discussion

The distribution and resistance patterns of BSI-causing pathogens vary according to time, geographical location, environment, population, and healthcare expenditure [18]. Knowledge of the baseline microbial resistance profile concerned with the hospital prevents irrational use of antibiotics in that hospital, thus helping progress a step forward in limiting the spread of antibiotic resistance. Globally, antimicrobial-resistant bacteria (ARB) have been recognized as a threat to public health. Though detection of the causing microbial by using molecular techniques has been proven suboptimal, blood culture remains the gold standard and first-line tool in the pathogen diagnostics of BSIs and provides clinically relevant information concerning the identity and analysis of microorganisms with their susceptibility to antibiotics [19].

The present study was a retrospective cross-sectional study from November 2023 to October 2024, conducted in the Department of Microbiology, Dhaka Medical College, to analyze the positive blood culture isolates from patients with BSI. In this study, 12.4% of the isolated organisms were gram-positive, and GNB were 87.6%; these observations were comparable to those studies conducted by Prakash *et al* in Nepal, Mia *et al in* Bangladesh, and Alhumaid *et al* in Saudi Arabia [20,4,21]. In this current study, *Salmonella* spp accounted for 50.5% of GNB isolates from blood. Nasrin *et al.* also reported *Salmonella* as the prominent organism causing BSI in Bangladesh [22].

The isolation rate of GNB was slightly higher in male patients (57.2%) compared to females 42.8% which coincides with the outcome of Ejaz *et al* in Pakistan [13]. Although the present study does not show any significant differences (p-value 0.177) based on gender in Gram-negative isolates from blood, *E. coli* bacteremia was more frequent in women. The female anatomy and vaginal colonization by *E. coli* can be a risk factor for *E. coli* bacteremia from a urinary tract infection (UTI). In this observation, *Salmonella* spp was more prevalent in males, which agrees with the study conducted by Bhumbla *et al.* [23]. The variation in male-female proportion in this study could be attributed to factors such as the male population being more involved in outdoor activities in our context, exposing them to infection.

The age-specific prevalence of different bacterial pathogens reflects variations in infection patterns among distinct demographic groups. In age interval analysis, *Acinetobacter* spp was more prevalent in the < 18 years age group, suggesting higher vulnerability in these age groups due to factors such as immature immune systems, reduced antimicrobial activity by neutrophils and macrophages, reduced antigen presentation by dendritic cells, decreased NK cell killing, and compromised acquired lymphocyte activities in younger individuals [24]. Similarly, *Klebsiella* spp and *Pseudomonas* spp were most frequent in the < 18 years age group, likely due to increased susceptibility to hospital-acquired infections among pediatric patients due to their immature immunity, exposure to invasive procedures, and the multidrug-resistant organisms in the hospital environment. [9,25]. In contrast, *Salmonella* spp were more occurred (49.2%) in the 19–30 years age group, Prakash *et al*. from India also found the highest *Salmonella* spp isolation rate in the age group between 16–30 years (54.10%) [20].

The relationship between the distribution of bacterial isolates and hospital wards reveals a significant association, as indicated by the Chi-Square test (p < 0.001). *Salmonella* spp was predominantly isolated from OPD (76.8%), likely reflecting community-acquired infections. A study from Pakistan reflected similar trends, with *Salmonella* causing a significant proportion of bloodstream infections in outpatient settings [26].

In the ICU predominant organism was *Acinetobacter* spp, this is consistent with the findings of Saharman *et al*., where a significant burden of *Acinetobacter*, was observed in the intensive care unit setting [27]. Hospital-associated infections were dominated by *Acinetobacter* spp and *Pseudomonas* spp, particularly in ICUs. In contrast, a study by Mathur *et al*. reported *Klebsiella* spp as the most frequently identified pathogens among bloodstream infections in the ICU in India [28].

Upon exploring collective antimicrobial resistance pattern in GNB except *Salmonella* spp, the highest resistance were observed to ceftazidime (94.1%), followed by piperacillin-tazobactam (89.2%), and ciprofloxacin (87.7%), which were concordant to that reported by Parajuli *et al.* in Nepal [29]. In the current study, *Acinetobacter* spp and *Pseudomonas* spp demonstrated frequent resistance to third-generation cephalosporins, which were in agreement with the study by Nesa *et al.* and Sharmin *et al.* from Bangladesh, reported that 93.5% to 100.0% *Acinetobacter baumannii* were resistant to the extended spectrum of cephalosporins [30,31]. Saha *et al*. reported similar finding that *Pseudomonas* spp were highly resistant to ceftazidime (82.3%) [32]. On the contrary, *Pseudomonas* spp showed 60.9% carbapenem resistance in this study, whereas AlBahrani *et al.* found only 9% resistance in Saudi Arabia. These variations in the susceptibility rates may be associated with differences in antibiotic use in different geographical areas and hospital settings [33].

*E. coli* showed high resistance to ciprofloxacin, piperacillin-tazobactam, and third generatin cephalosporin group of drugs. Additionally, *Klebsiella spp.* exhibited high resistance to ceftazidime, followed by ceftriaxone and piperacillin-tazobactam; those findings have similarities with a study conducted in Bangladesh by Akter *et al*. [34]. Overall carbapenem resistance rate was 63.2%, which is slightly higher than Aminul *et al.* reported from Bangladesh [35]. Carbapenem resistance in Enterobacterales has been particularly worrisome in South Asian settings, attributed to the widespread dissemination of carbapenemase-producing strains [36].

In this current study, among the total 374 isolated organisms, 201 (53.7%) were identified as MDR and 58 (15.5%) as XDR. *Salmonella* spp (40.7%) demonstrated the lowest MDR rate, with no isolates classified as XDR. This was supported by previous studies conducted by Mina *et al*., where *Salmonella* spp has shown variable susceptibility to ciprofloxacin but limited resistance to other groups of antibiotics [37]. *Acinetobacter* spp exhibited the highest MDR (75.4%) and XDR (42%) rates, which were consistent with study reports by Banerjee *et al.* and Sharmin *et al.,* where *Acinetobacter* spp has

 

## PLOS Global Public Health

shown a high resistance rate to multiple antibiotic classes, including carbapenems [38,31]. These findings highlight the alarming prevalence of antimicrobial resistance (AMR) in hospital settings, which poses a grave threat to public health, particularly in low-income countries in South Asia, including Bangladesh.

The present study also revealed significant disparities in MDR and XDR prevalence across hospital units. The ICU had the highest proportion of MDR and XDR isolates, followed by the inpatient department (IPD); this trend is consistent with studies conducted by Van An *et al.* in Vietnam [39]. ICU settings are a hotspot for nosocomial infection with AMR pathogens, due to the high usage of broad-spectrum antibiotics, invasive procedures, and prolonged hospital stays. The lowest MDR prevalence was observed in the outpatient department (OPD), likely reflecting reduced exposure to hospital-acquired infections and limited prior antibiotic use.

The rise of antimicrobial resistance (AMR) and bloodstream infections (BSIs) in Bangladesh underscores the urgent need for effective Antimicrobial Stewardship Programs (ASP) and infection control practices. Effective diagnostics, infection prevention, and responsible antibiotic use are critical to combating this growing threat. The study findings provide insight into the recent antimicrobial profile, which will strengthen the antimicrobial stewardship programme and guide to formulate an effective antibiogram in this tertiary healthcare setting.

Overall, this study was a single institution-based retrospective observational study with some limitations. CLSI 2022 breakpoints were used to interpret the antibiotic susceptibility testing, whereas some changes in zone diameters were adopted by CLSI afterwards, which were not considered in this study. Patient outcome, mortality, morbidity rates, and other risk factors were not measured in this study due to a lack of data. Advanced molecular methods can provide deep insight into the isolation and resistance profile, which were not considered in this study. Therefore, the findings must be interpreted with caution, and further studies should be conducted on a larger sample involving several hospitals from different geographical areas.

## Conclusion

GNB isolated from the blood sample had a slight predominance of male gender. *Salmonella spp.* were prominent isolates in OPD, whereas *Acinetobacter* spp and *Pseudomonas* spp were more prevalent in ICU and IPD settings. Most of the isolates show high resistance to cephalosporin, piperacillin-tazobactam, and ciprofloxacin. This current study also highlights the high prevalence of MDR and XDR Gram-negative pathogens in bloodstream infections. The ICU remains the most affected setting, and *Acinetobacter* spp emerges as a key pathogen of concern. Regular updates on the epidemiology of BSIs, including geographic and climate-driven variations in antibiotic resistance patterns, are essential for antibiotic stewardship that ensures timely and effective treatment.

## Supporting information

**S1 Data. This file contains all the data collected from Microbiology laboratory records regarding gram-negative bacterial growth and antimicrobial susceptibility testing results of Dhaka Medical College, which were used for statistical analysis in this study as outlined in the methods section of the main manuscript.**
(SAV)

## Acknowledgments

We acknowledged the staff of the Microbiology Department, Dhaka Medical College, for their contributions.

## Author contributions

**Conceptualization:** Nusrat Noor Tanni, Rubaiya Binte Kabir, Kakali Halder, Mahbuba Chowdhury, Sazzad Bin Shahid.
**Formal analysis:** Nusrat Noor Tanni, Maherun Nesa, Farjana Binte Habib, Rozina Aktar Zahan, Mahbuba Chowdhury.

**Investigation:** Nusrat Noor Tanni, Md. Asaduzzaman, Avizit Sarker, Noor E Jannat Tania, Azmeri Haque, Umme Saoda.

**Methodology:** Nusrat Noor Tanni, Maherun Nesa, Rubaiya Binte Kabir, Farjana Binte Habib, Md. Faizur Rahman, Azmeri Haque, Umme Saoda, Nadira Akter.

**Project administration:** Nusrat Noor Tanni.

**Supervision:** Sazzad Bin Shahid.

**Writing – original draft:** Nusrat Noor Tanni, Mahbuba Chowdhury, Sazzad Bin Shahid.

**Writing – review & editing:** Nusrat Noor Tanni, Maherun Nesa, Rubaiya Binte Kabir, Farjana Binte Habib, Md. Asaduzzaman, Avizit Sarker, Md. Faizur Rahman, Noor E Jannat Tania, Kakali Halder, Nadira Akter, Rozina Aktar Zahan, Mahbuba Chowdhury.

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
