## [Decision Letter · Decision Letter 0]

8 Aug 2025

PGPH-D-25-01638

Distribution, epidemiology, and antimicrobial resistance pattern of gram-negative bacteria isolated from blood: a retrospective study in a tertiary care hospital, Dhaka, Bangladesh

Dear Authors

Thank you for submitting your manuscript to PLOS Global Public Health. After careful consideration, we feel that it has merit but does not fully meet PLOS Global Public Health’s publication criteria as it currently stands. Therefore, we invite you to submit a revised version of the manuscript that addresses the points raised during the review process.

We look forward to receiving your revised manuscript.

Kind regards,

Mohamed Salah Abbassi

Academic Editor

Journal Requirements:

1. Please provide additional details regarding participant consent. In the ethics statement in the Methods and online submission information, please ensure that you have specified (1) whether consent was informed and (2) what type you obtained (for instance, written or verbal, and if verbal, how it was documented and witnessed). If your study included minors, state whether you obtained consent from parents or guardians. If the need for consent was waived by the ethics committee, please include this information.

If you are reporting a retrospective study of medical records or archived samples, please ensure that you have discussed whether all data were fully anonymized before you accessed them and/or whether the IRB or ethics committee waived the requirement for informed consent. If patients provided informed written consent to have data from their medical records used in research, please include this information."

-DOI: 10.7759/cureus.11000

-https://doi.org/10.1016/j.mehy.2020.109861

In your revision ensure you cite all your sources (including your own works), and quote or rephrase any duplicated text outside the methods section. Further consideration is dependent on these concerns being addressed.

3. We have amended your Competing Interest statement to comply with journal style. We kindly ask that you double check the statement and let us know if anything is incorrect.

4. We note that your Data Availability Statement is currently as follows: [All the data related to this manuscript have been uploaded as Supporting Information files.]

Additional Editor Comments (if provided):

Dear Authors

Thank you for submitting your article in PGPH. Reviewer has submitted his comments; wile he find your article merit publication, he advise for'Major Revision'.

Please re-submit a corrected copy within 45 days, and highlight all corrections by red color. In addition please respond to all reviewers comments one by one.

Sincerely

Reviewer report

Distribution, epidemiology, and antimicrobial resistance pattern of gram-negative bacteria isolated from blood: a retrospective study in a tertiary care hospital, Dhaka, Bangladesh

• What are the main claims of the paper and how significant are they for the discipline?

Although the claim is raised in the background of the text as highlighted below, it is important that authors should highlight in the abstract section.

Moreover, authors should draw further attention to the scarcity of epidemiological data in the study population relevant to the study settings and explain its importance by examining AMR trends/infection prevalence in the study setting.

it is well acknowledged that AMR 77 is a worldwide concern that mostly affects low-income countries. Scrutiny of microorganism 78 distribution, BSI epidemiology, changing antibiotic resistance rates, and demographics are 79 required to support appropriate therapy. This study aimed to ascertain the prevalence, distribution, 80 and antimicrobial resistance patterns with MDR, and XDR status, of commonly isolated GNB by 81 age, sex, and hospital units through the analysis of the blood cultures of patients at Dhaka Medical 82 College Hospital (DMCH), a tertiary care hospital in Dhaka, Bangladesh.

Material and Methods:

Although one of the objectives of the study is to provide prevalence data, there is no clear evidence of sample size calculation or indication of the number of Blood isolates considered for the analysis.

Bacterial blood culture yielded 427 isolates during the study period. This section comes from the Results section. However, this information needs to be provided at the beginning, along with the number of isolated collections.

Further, there is no justification for the selected antibiotics for AST screening. Is it based on the local guidelines?

• Do the data and analyses fully support the claims? If not, what other evidence is required?

Results, discussion and conclusion;

Authors have considered an appropriate methodology and highlighted the study's limitations, but it is not apparent what is significant to the discipline.

The majority of the study area's findings are similar to those previously published; therefore, it is challenging to determine the significance of AMR trends/prevalence patterns in the study location.

The study: Distribution, epidemiology, and antimicrobial resistance pattern of gram-negative bacteria isolated from blood: a retrospective study in a tertiary care hospital, Dhaka, Bangladesh has provided the original insight in the study setting, but it overall lacks the significance of the data to the study field.

Authors can draw from published literature to strengthen and justify the claim that they are raising in the paper.

Reviewers' comments:

Reviewer's Responses to Questions

**Comments to the Author**

1. Does this manuscript meet PLOS Global Public Health’s publication criteria?

Reviewer #1: Partly

2. Has the statistical analysis been performed appropriately and rigorously?

Reviewer #1: No

3. Have the authors made all data underlying the findings in their manuscript fully available (please refer to the Data Availability Statement at the start of the manuscript PDF file)?

Reviewer #1: Yes

4. Is the manuscript presented in an intelligible fashion and written in standard English?

Reviewer #1: Yes

Reviewer #1: Distribution, epidemiology, and antimicrobial resistance pattern of gram-negative bacteria isolated from blood: a retrospective study in a tertiary care hospital, Dhaka, Bangladesh

• What are the main claims of the paper and how significant are they for the discipline?

Although the claim is raised in the background of the text as highlighted below, it is important that authors should highlight in the abstract section.

Moreover, authors should draw further attention to the scarcity of epidemiological data in the study population relevant to the study settings and explain its importance by examining AMR trends/infection prevalence in the study setting.

it is well acknowledged that AMR 77 is a worldwide concern that mostly affects low-income countries. Scrutiny of microorganism 78 distribution, BSI epidemiology, changing antibiotic resistance rates, and demographics are 79 required to support appropriate therapy. This study aimed to ascertain the prevalence, distribution, 80 and antimicrobial resistance patterns with MDR, and XDR status, of commonly isolated GNB by 81 age, sex, and hospital units through the analysis of the blood cultures of patients at Dhaka Medical 82 College Hospital (DMCH), a tertiary care hospital in Dhaka, Bangladesh.

Material and Methods:

Although one of the objectives of the study is to provide prevalence data, there is no clear evidence of sample size calculation or indication of the number of Blood isolates considered for the analysis.

Bacterial blood culture yielded 427 isolates during the study period. This section comes from the Results section. However, this information needs to be provided at the beginning, along with the number of isolated collections.

Further, there is no justification for the selected antibiotics for AST screening. Is it based on the local guidelines?

• Do the data and analyses fully support the claims? If not, what other evidence is required?

Results, discussion and conclusion;

Authors have considered an appropriate methodology and highlighted the study's limitations, but it is not apparent what is significant to the discipline.

The majority of the study area's findings are similar to those previously published; therefore, it is challenging to determine the significance of AMR trends/prevalence patterns in the study location.

The study: Distribution, epidemiology, and antimicrobial resistance pattern of gram-negative bacteria isolated from blood: a retrospective study in a tertiary care hospital, Dhaka, Bangladesh has provided the original insight in the study setting, but it overall lacks the significance of the data to the study field.

Authors can draw from published literature to strengthen and justify the claim that they are raising in the paper.

**Do you want your identity to be public for this peer review?** For information about this choice, including consent withdrawal, please see our Privacy Policy

Reviewer #1: **Yes: ** Shivanthi Samarasinghe

---

## [Editor Report · Decision Letter 1]

15 Oct 2025

Distribution, epidemiology, and antimicrobial resistance pattern of gram-negative bacteria isolated from blood: a retrospective study in a tertiary care hospital, Dhaka, Bangladesh

PGPH-D-25-01638R1

Dear Dr Tanni,

We are pleased to inform you that your manuscript 'Distribution, epidemiology, and antimicrobial resistance pattern of gram-negative bacteria isolated from blood: a retrospective study in a tertiary care hospital, Dhaka, Bangladesh' has been provisionally accepted for publication in PLOS Global Public Health.

Best regards,

Mohamed Salah Abbassi

Academic Editor

Authors have perfectly responded to the previous comments of the reviewers. I am satisfied and the article seems better now. I believe that the article can be accepted.

Sincerely